

# Coastal vulnerability assessment: through regional to local downscaling of wave characteristics along the Bay of Lalzit (Albania)

Francesco De Leo[1], Giovanni Besio[1], Guido Zolezzi[2], and Marco Bezzi[2]

[1]Dept. of Civil, Chemical and Environmental Engineering - University of Genoa, Genoa, 16145, Italy
[2]UNESCO Chair in Engineering for Human and Sustainable Development - Dept. of Civil, Environmental and Mechanical Engineering - University of Trento, Trento, 38123, Italy

*Correspondence to:* Francesco De Leo (francesco.deleo@edu.unige.it)

**Abstract.**

Coastal vulnerability is evaluated against inundation risk triggered by waves run-up, through the employment of coastal vulnerability indexes (referred to as "CVI") introduced by Bosom García and Jiménez Quintana (2011). $CVI$ are assessed through different wave climate characterizations, referring to regional (offshore wave climate) or local (near-shore wave climate) scale. The study is set along the Lalzit Bay, a coastal area nearby Durrës (Albania). The analysis reveals that the results vary due to uncertainties inherent in the run-up estimation, showing that the computational procedure should be developed by taking into account detailed information about local wave climate, especially concerning seasonal behaviour and fluctuations. Different approaches in choosing wave characteristics for run-up estimation affect significantly the estimate of shoreline vulnerability. The analysis also shows the feasibility and challenges of applying $CVI$ estimates in contexts characterized by limited data availability, through targeted field measurements of the coast geomorphology and an overall understanding of the recent coastal dynamics and related controlling factors.

## 1 Introduction

Coastal zones are often characterized by a fragile equilibrium, being subjected to hydro-geomorphic processes that change their shape over time and space, and are as well under stress due to the presence of conflicting human activities (Kamphuis, 2010). Moreover, these areas have a huge socio-economic value, which has often triggered their high exploitation in the last decades: coastal population is constantly increasing, together with maritime commerce and coastal tourism (Neumann et al., 2015). This implies enhanced anthropogenic pressures, which challenge their sustainable management and preservation.

The present paper focuses on extreme natural storm events and on their impact on coastal vulnerability within such complex framework. As clearly specified by the Integrated Protocol on Coastal Zone Management ("ICZM"), the effect of storms should be embedded into coastal zone territorial plans and policies, yielding coastal vulnerability assessment (UNEP, 2008). Efficient assessment and decision support tools are required, providing easily accessible information for decision-makers.





Coastal Vulnerability Indexes represent a viable assessment option, because they are helpful to classify the shorelines in relation to their vulnerability towards extreme events, such as storms induced inundation and erosion.

These indexes usually take into account the long term wave statistics and the geomorphology of the beaches to evaluate the level of risk they are exposed to. The estimate of the environmental risk, coupled with the evaluation of the existing anthropic pressure (economic and industrial activities) leads to vulnerability maps. Different approaches to compute $CVI$ have been so far proposed, which differently combine environmental and socio-economic relevant variables (Soukissian et al., 2010; Di Paola et al., 2014; Fitton et al., 2016; Satta et al., 2016; Armaroli and Duo, 2017; Ciccarelli et al., 2017; Óscar Ferreira et al., 2017; Ferreira Silva et al., 2017; Montreuil et al., 2017; Narra et al., 2017; Mavromatidi et al., 2018, among others). A methodological issue of particular concern is related to the computation of wave climate characteristics suitable for calculating values of the $CVI$ that are of management significance. This can be illustrated by referring to the practical procedure proposed by Bosom García and Jiménez Quintana (2011), to assess coastal vulnerability to inundation. The procedure foresees to compute the long term run-up values, starting from the ones evaluated through the model of Stockdon et al. (2006), and then combining it with the berm or dune heights of a shore to achieve its run-up vulnerability. However, Stockdon et al. (2006) formulation intrinsically leads to a very conservative result, as it provides a value linked to the 2% probability of exceedance: this means that, for given wave and beach characteristics, the computed run-up is not the one most likely occurring, but one of the highest possibly observed within a hypothetical series of records. Therefore, evaluation of high return period run-up values could lead to highly overestimated coastal vulnerability, as the return period is closely tied to the probability of non exceedance. Conversely, the model of Stockdon et al. (2006) has shown to provide more reliable run-up estimates if the input wave parameters are provided in the near-shore region at a depth of 10 m, instead of referring to deep-water values as the original formulation suggests (Sancho-García et al., 2012). This requires to change the scale of the wave climate characterization, moving from a national/regional scale to a more detailed local scale in order to obtain reliable estimates for coastal inundation. Such downscaling of the risk analysis implies that wave conditions have to be first propagated towards the shore to evaluate shallow water parameters and afterwards to compute run-up accordingly, leading to more reliable run-up *expected value* estimate in the case of extreme events (Di Risio et al., 2017).

The main goal of the present paper is to quantify differences in assessing coastal vulnerability to inundation when using a regional and a local (near-shore) characterization of the wave climate, and in consideration of its seasonality. The study refers to the bay of Lalzit, immediately north of the city of Durrës (Albania, see Fig. 1). The focus on such rapidly developing context also allows to discuss the potential implications of coastal vulnerability assessment when decision-making requires to be highly adaptive and when data availability is scarce. Preliminary studies on the characteristic wave climate characterized its seasonal directional frames. Building such information, it has been possible to compute seasonal vulnerability indexes to be then compared with the offshore omni-directional ones. Such seasonal risk approach is particularly relevant because it could highlight the critical issues related to coastal zone management when the littoral use and exploitation change drastically between different seasons, and represents an additional novelty of the present work.




$CVI$ assessment was performed through the Bosom García and Jiménez Quintana (2011) procedure, referring both to offshore and near-shore wave data, to evaluate variations in shoreline vulnerability depending on the employed spatial (regional/local) and temporal scale (extreme events/seasonal/directional).

The paper is organized as follows: in Sect. 2 we present the indexes computation procedure, along with the investigation
area and the data used; in Sect. 3 we show results of coastal vulnerability using wave dataset at regional and near-shore scales; in Sect. 4, results are presented and possible future developments and improvements are discussed.

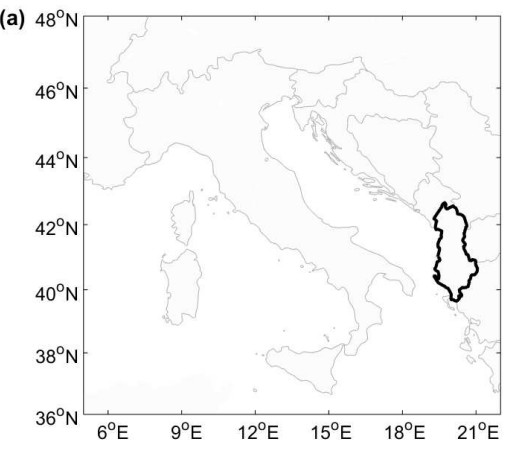

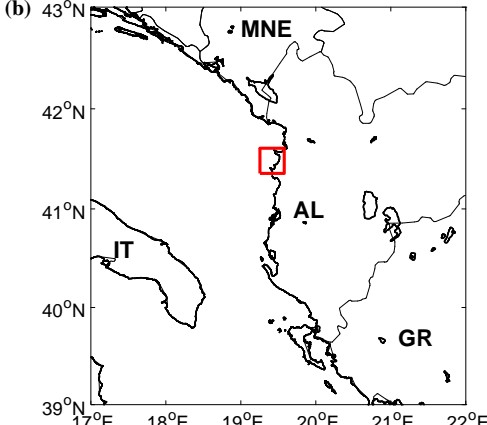

**Figure 1.** Map showing the area under investigation. A) Location of Albania in South-East Europe and B) the bay of Lalzit underlined within the red frame

## 2   Data & Methods

The vulnerability assessment is part of a wider research project, aimed at evaluating and quantifying the ongoing coastal erosion affecting the Lalzit bay area. This phenomenon is due to different causes, concerning the abrupt changes occurred in Albania
after the fall of the communist regime that triggered very fast morphological changes either in the watersheds and on the coast (De Leo et al., 2017). In order to collect all the required data, a two weeks field campaign was performed during the month of July 2015.

### 2.1   Study area: Lalzit Bay, Albania

The Lalzit Bay is included between two capes, and can therefore be considered as an independent physiographic unit; it is
possible to focus on the processes affecting this coastline independently from those characterizing the nearby physiographic units. A physiographic unit is indeed defined as a portion of shoreline with coherent characteristics in terms of natural coastal processes and of land use, which can thus be studied independently from neighbouring shores (UNEP, 2008). Specifically, in




the Lalzit bay area we identified four main interdependent geophysical and anthropogenic processes that control the dynamics and evolution of the coastal zone (see Fig. 2). The strong interaction among such processes has been identified by De Leo et al. (2017) as the main mechanism of a massive retreat of the coastline over the past thirty years.

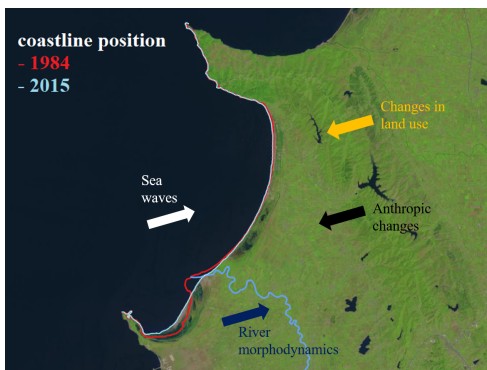

**Figure 2.** Conceptual diagram of the main geophysical and anthropogenic controls on the coastal dynamics of the Lalzit bay. The red and light blue lines show the temporal variation of the coastline position.

## 2.2 Field measurements

Field activities were aimed at collecting the minimum required data to investigate the relevant processes affecting coastal dynamics presented in Fig. 2. The geomorphology of the beaches along the bay was characterized through sixteen sections (e.g. Fig. 3) crossing the shoreline, nearly spaced every kilometre along almost twenty km of the bay length (from sec -4, south, to sec 11, north, see Fig. 4A). We recorded the cross-shore section elevation at topographically relevant locations, in correspondence of the main slope changes, with particular attention to the submerged bar system. This allowed to assess the

cross-shore sections shape, their berm height and the overall cross-shore profile mean slope (Fig. 3). Moreover, we collected different sand samples along every section to characterize their grain size distribution. Sediment samples were taken at selected locations along each section. Every sand sample was analysed through a multi-filter sieve, to assess the weight percentages of sand in each size class, thus building the grading curve. The obtained data were then post-processed by using the software Gradistat (Blott and Pye, 2001), further evaluating the median grain size ($d_{50}$) for every sampled location. As the resulting

values of $d_{50}$ were not significantly varying along each cross-shore profile, we chose to use those characterizing the water edge foreshore as the representative ones of each section.

Results of the grain size surveys are summarized in Fig. 4. The mean grain size ($d_{50}$) happens to be quite homogeneous among all the sections (Fig. 4B), and the granulometry of the bay can be considered representative of a "medium sand", according to the classification of Wentworth (1922). The only exception is represented by the section next to the Rodoni cape,

which is close to a rocky promontory and is therefore characterized by coarser sediments. On the other hand, cross-shore mean slopes ($\beta_f$) and berm heights ($B_h$) are more variable along the coast, with steeper sections being characterized by lower berms and vice-versa (Fig. 4C, D).





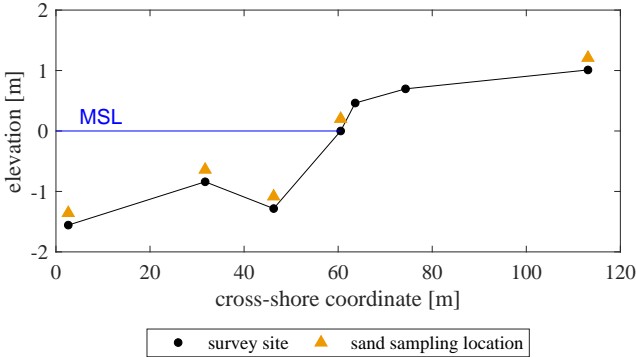

**Figure 3.** Typical cross-shore profile along the Lazlit Bay (example of section 2, Figure 4A). It is possible to note the presence of the submerged bar some tens of meters away from the coastline

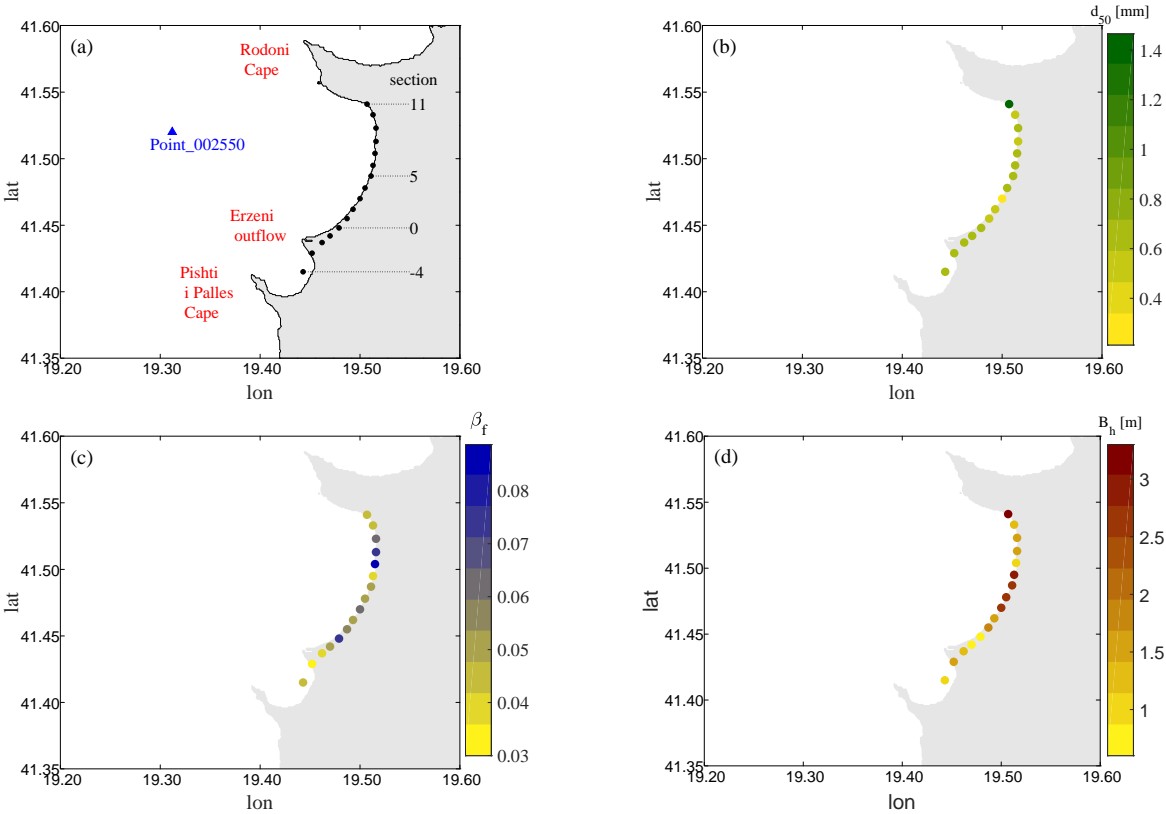

**Figure 4.** A) Sampling locations for beach sections (from -4 to 11, from south to north), Point_002550 represents DICCA wave hindcast. Spatially distributed values of: B) Median grain size ($d_{50}$); C) Cross-shore mean slope ($\beta_f$); D) Berm height ($B_h$)





## 2.3 Coastal Vulnerability Indexes (CVI)

$CVI$ are meant to quantify the vulnerability of a coast toward extreme inundation events. For the investigated beach section (or length of shore), a long term statistical computation for the run-up is required, leading to an intermediate dimensionless variable $IV$ ("Inundation Vulnerability"), defined as follows:

$$IV = \frac{Ru_{2\%}}{B_h} \tag{1}$$

where $B_h$ and $Ru_{2\%}$ are the beach berm or dune height and the long term run-up respectively. For each section, the $IV$ value is then evaluated within a given range, obtained by setting two boundary values:

$$
\begin{cases}
IV_{min} = \dfrac{Ru_{2\%}}{2 Ru_{2\%}} & \Rightarrow Ru_{2\%} = 0.5\ B_h \\[2em]
IV_{max} = \dfrac{Ru_{2\%}}{Ru_{2\%} - 2} & \Rightarrow Ru_{2\%} = 2 + B_h
\end{cases}
\tag{2}
$$

It can be noticed that the minimum and the maximum vulnerability levels have a clear physical meaning, being explanatory of the cases where the run-up is either half ($IV_{min}$) of or two meters higher than ($IV_{max}$) the berm height. This interval is then scaled to a range from 0 to 1, grouped in five classes of equally spaced vulnerability levels ("very low", "low", "medium", "high", "very high") as reported in Table 1.

| IV | 0 - 0.2 | 0.2 - 0.4 | 0.4 - 0.6 | 0.6 - 0.8 | 0.8 - 1.0 |
|---|---|---|---|---|---|
| CVI | very low | low | medium | high | very high |

**Table 1.** Vulnerability levels assessment due to the $IV$ variable

## 2.4 Wave data and run-up

The assessment of $CVI$ first requires to compute the long term run-up statistics. Regardless the reference model, run-up computation always implies to combine informations about both characteristic wave climate and morphology of a shore (Battjes, 1971; Holman, 1986; Mase, 1989, among others). As regards the wave data, we referred to the hindcast provided by the Department of Civil, Chemical and Environmental Engineering of the University of Genoa ("DICCA", dicca.unige.it/meteocean/hindcast). The hindcast is defined all over the Mediterranean sea from 1979 to 2016 with a 0.1° resolution both in longitude and latitude, one hourly sampled, and it is based on NCEP Climate Forecast System Reanalysis ("CFSR"), for the period from January 1979 to December 2010 and CFSv2 for the period from January 2011 to December 2016 (Mentaschi et al., 2013). The DICCA hindcast was widely validated (Mentaschi et al., 2015), and, being densely defined over a large time period, it helps to perform reliable long-term statistical computations (Coles and Pericchi, 2003). The location we referred to for this study is shown in Fig. 4A (Point_002550), whereas data about the shore geomorphology were collected as explained in Sect. 2.2.



Run-up is therefore computed according to Stockdon et al. (2006) model, as follows:

$$Ru_{2\%} = 1.1 \left\{ 0.35 \, \beta_f \, \sqrt{H_0 L_0} + \frac{\left[ H_0 L_0 \left( 0.563 \, \beta_f^2 + 0.004 \right) \right]^{0.5}}{2} \right\} \tag{3}$$

where $\beta_f$ stands for the mean slope of the beach, $H_0$ and $L_0$ refer to deep water wave height and length respectively.

## 2.5  Extreme Value Analysis (EVA)

When dealing with run-up estimation, if the data linked to the shore characteristics can be well defined, more uncertainties grow up when trying to empirically parametrize exceptional phenomena (extreme events), of which run-up can be considered as an instance. For this reason we tested two different approaches for the estimation of extreme run-up values.

First, in the frame of a regional analysis, we considered the deep-water data as defined in Point_002550, selecting the annual maxima sea storms from the wave dataset and evaluating the annual maxima run-ups through Eq. (3). This resulted in a 38

extreme run-up dataset for each of the sixteen sections. Every dataset was then modelled through a GEV distribution (Coles et al., 2001), in order to carry out the long term design of run-up values; the validity of the distribution was always proved through the Kolmogorov-Smirnov and the Anderson-Darling parametric tests for every dataset (Massey Jr, 1951; Anderson and Darling, 1954). Given the distributions, we set two target return periods, literally 50 yr and 200 yr, and further computed the resulting run-ups for every section in both cases. This allowed to quantify how CVI estimation could be affected by differently

conservative approaches.

Afterwards, we switched from a regional to a locale scale: in this case, EVA were performed directly over the extreme sea storms wave parameters, to assess the 50 yr and the 200 yr waves. We thus propagated the target waves in front of each section, computing afterwards the long term run-up values. Here, as the wave climate shows a clear seasonal dependence with respect to the average incident waves direction, we splitted the initial wave dataset according to two meaningful directional fetches;

this choice involved two important consequences.

First, when performing directional analysis, referring return periods for each of the identified sectors have to be carefully assigned, as demonstrated by Forristall (2004):

$$\begin{cases} F_o = \prod_1^N F_i \\ 1 - \dfrac{1}{\lambda_o T_{R_o}} = \prod_1^N \left( 1 - \dfrac{1}{\lambda_i T_{R_i}} \right) \end{cases} \tag{4}$$

being $F$ the probability of non exceedance, $\lambda$ the yearly number of extreme events, $T_R$ the significant return period; sub-

scripts $o$ and $i$ stand for omnidirectional and the $i^{-th}$ directional pattern, respectively. The $F_i$ probabilities are fixed in order to obtain $N$ equal values whose product gives $F_o$. This precaution is due to the fact that, when referring to an omnidirectional analysis, a higher number of events is expected compared to the case of a directional analysis, thus a higher probability of non exceedance (see Eq. (4)). Multiplying the probabilities related to the identified sectors ensures a coherent computational procedure; we therefore previously fixed the referring non-exceedance probability for each sector, in order to get an overall

value equal to that of the omnidirectional analysis (given the referring $T_{R_o}$).





Moreover, referring to a subset of the whole dataset implies a lower amount of data to deal with. In order to overcome this drawback, we used the Peaks Over Threshold (POT) approach to come up with long term wave height estimates, imposing a threshold value of 3 m and a minimum inter-event duration of 24hr; threshold values were set to get datasets characterized by events being independents and identically distributed, as specified by Lang et al. (1999). Probabilities obtained with Eq. (4)

lead to the long term significant wave heights; in this case we adopted a Weibull distribution applied to the exceedances of the given threshold, as specified in DNV (2010), testing as well the suitability of the distribution as performed for the regional analysis. To completely characterize the target waves (to be downscaled at a later time in the near-shore zone), we linked the peak periods to the computed significant wave heights according to the empirical formula of Callaghan et al. (2008); the mean directions were assessed instead due to the particular waves climate of the area. Resulting values were set as inputs to the

SWAN model (Booij et al., 2003), allowing to get the parameters at a depth of ten meters in front of each of the investigated sections. The obtained wave parameters were then used to compute the 10 m depth run-up for both the considered return periods. As regards the bathymetry of the bay, we referred both to the ETOPO1 dataset (www.ngdc.noaa.gov) and a nautical chart of the Italian Hydrographic Institute (www.marina.difesa.it).

It is worth mentioning that the return period of a forcing variate is not necessarily equal to the return period of the outcomes

(Hawkes et al., 2002); as an instance, a given return period wave may not lead to the corresponding return period run-up. Anyway, Garrity et al. (2007) demonstrated that this approach can still leads to satisfactory results, and it has already been adopted within previous studies (Vitousek et al., 2008).

## 3 Results

Once we computed the long term run-ups, we evaluated the resulting $CVI$ according to the morphology of the testing locations.

Since results are punctual (e.g. one index for each of the sixteen sampling locations). We linearly interpolated the $IV$ values within hypothetical intermediate sections, in order to get a more meaningful overview about the whole bay.

We initially referred to the regional scale; in this case, an omnidirectional analysis was performed, leading to two sets of results linked to the tested return periods. Secondly, we detailed our study to the local scale: in this case, we got two sets of results for every directional sector taken into account. We first present the $CVI$ obtained from the regional study.

### 25 3.1 Regional scale (offshore wave conditions)

At the regional scale the environmental inputs were the same for each section, being the wave characteristics defined in deep water (Point_002550, Fig. 4A); the differences in the run-up significant values were just due to different morphological characteristics of each cross-shore section (literally, the mean slope of the different beach profiles). This can be clearly noticed in Fig. 5: the empirical run-ups show the same distribution for every section, as their values are just rigidly translated of a quantity

that depends on the value of the section slope $\beta_f$ (see Eq. 3). From the curves in Fig. 5, the run-ups linked to 50 and 200 yr return period were extrapolated, and the inundation vulnerability indexes were accordingly computed, as explained in section



2.3. Results are shown in Fig. 6.

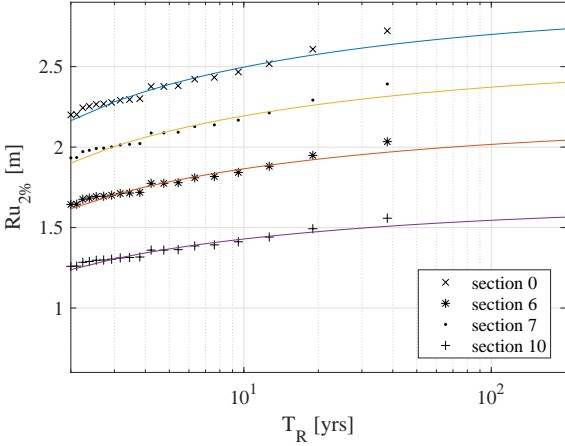

**Figure 5.** Return period curves for the run-up parameter; results are presented for just some of the cross-sections for the sake of clarity

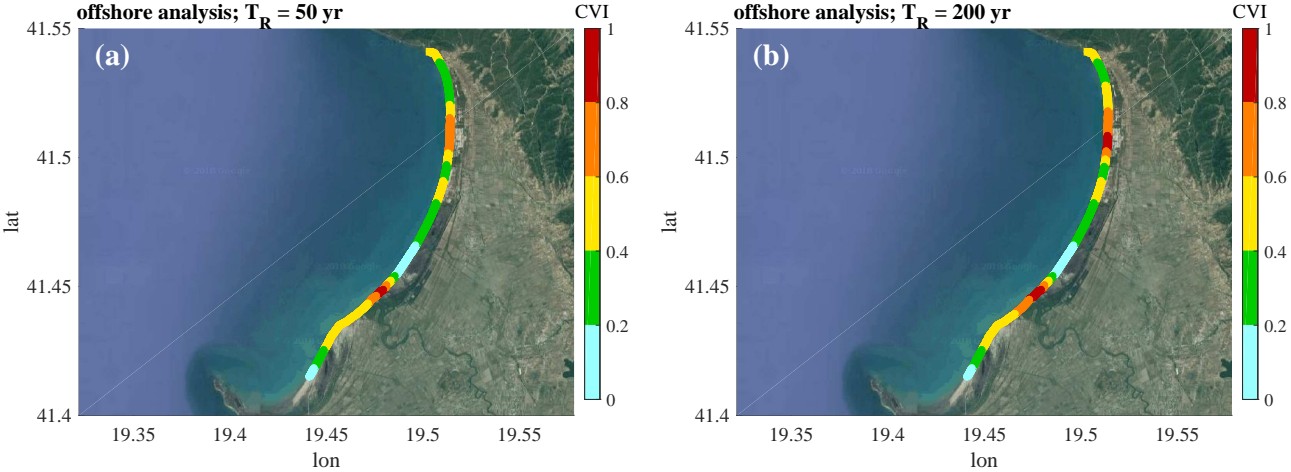

**Figure 6.** Run-up vulnerability indexes for the Lalzit Bay from the regional analysis, using deep water data: A) 50 yr return period; B) 200 yr return period





### 3.2 Local scale (nearshore wave conditions)

Evaluation of coastal vulnerability indexes has been carried out also by employing the propagated values of the wave climate at the local scale. It has to be remarked that, in this case, the mean cross-shore slope is not the only changing parameter between one section and another: as waves are propagated toward the shore in front of each of the investigated locations, they

5 are modified due to the occurring transformation processes, resulting in different wave characteristics (heights, lengths and incident directions) depending on the position of a section along the bay.

The first step to compute the indexes at a local scale is to characterize the wave climate. We mainly referred to the significant wave height, looking at the distribution of the waves incoming direction over different seasons. As shown in Fig. 7, the wave climate of the Lalzit bay is characterized by a strong seasonal behaviour: during the summer months, the prevalent incoming

10 direction happens to be W-NW, whereas for the other seasons waves mainly come from the S-SW direction. We therefore considered two directional sectors, 250-350° N and 190-250° N, being representative of summer and of the rest of the year, respectively.

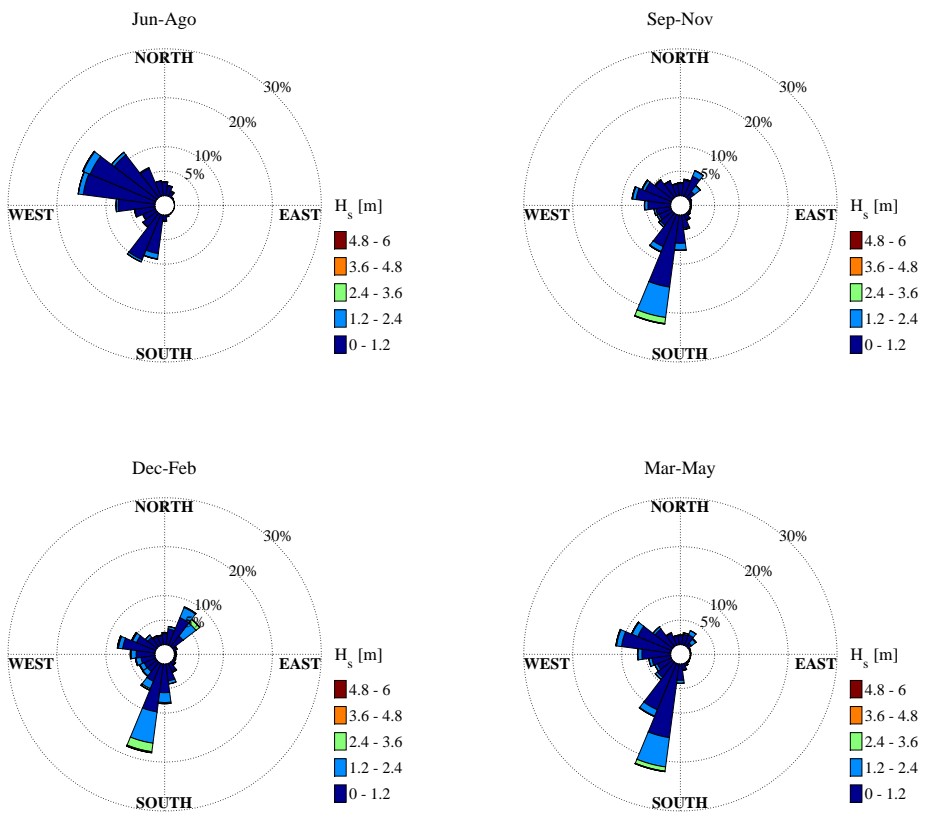

**Figure 7.** Seasonal roses of significant wave height for hindcast Point_002550.





Extreme events have been defined for each of the identified sectors, computing the resulting 50 yr and 200 yr return period wave heights. The target wave incoming direction for each sector was defined through a linear interpolation, in order to minimize the root mean square error with respect to the directions of the sea storms identified with the POT approach (see Fig. 8). Finally, for the wave periods, we referred to the empirical equation of Callaghan et al. (2008):

$$
\begin{cases}
E(T_p) = aH_s^b + cfH_s^{d+g} \\
a = 3.005; b = 0.543; c = 4.82; d = -0.332; f = 1.122; g = -0.039
\end{cases}
\tag{5}
$$

where $H$ is the target wave height computed through the POT approach, as previously explained; $a$, $b$, $c$, $f$, $d$ and $g$ are given parameters.

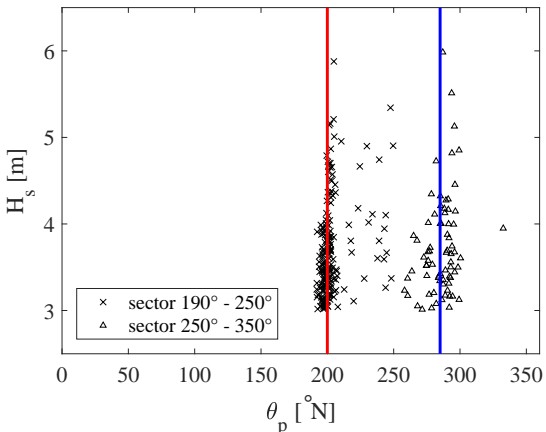

**Figure 8.** Directions of the extremes waves extracted through the Peaks Over Threshold procedure.

We therefore characterized the design wave for each of the identified directional sectors (W-NW and S-SW), defining its significant height, peak period and angle of attack. These parameters were set at a time as inputs of the wave propagation model, in order to get the shallow water wave parameters. The starting values are shown in Table 2. The inundation vulnerability indexes following the downscaled wave features are shown in Fig. 9 and 10.

| sector [°N] | $T_R$ [yrs] | $H_s$ [m] | $T_p$ [s] | $\theta_p$ [°N] |
|---|---|---|---|---|
| 190-250 | 50 | 6.49 | 11.0 | 200 |
|  | 200 | 7.01 | 11.3 | 200 |
| 250-350 | 50 | 6.64 | 11.1 | 285 |
|  | 200 | 7.17 | 11.4 | 285 |

**Table 2.** Design wave parameters for the seasonal directional sectors. Notations $T_R$ is the return period, $H_S$, $T_P$, and $\theta_P$ stand for wave height, period and incoming direction, respectively.



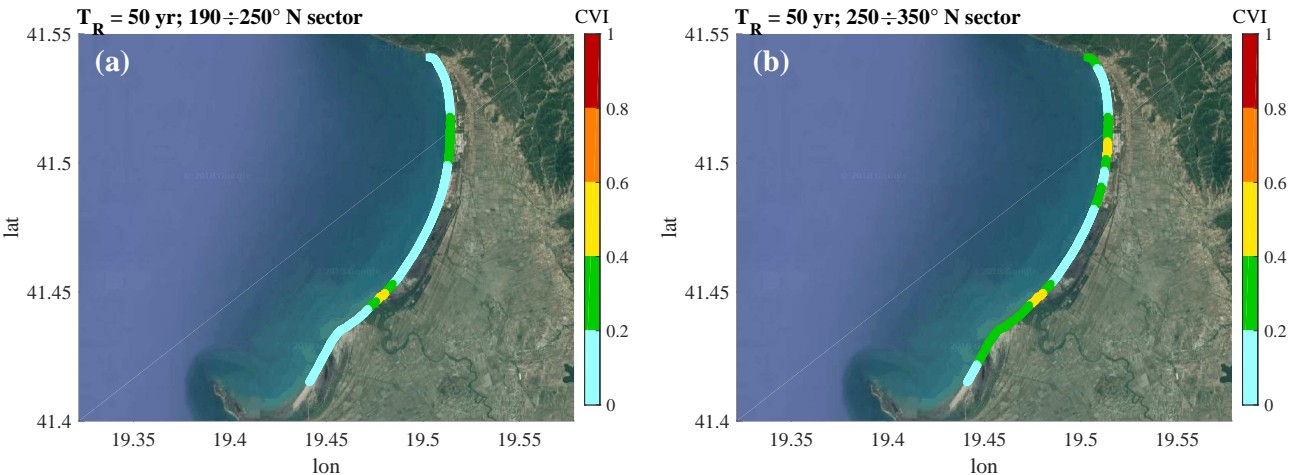

**Figure 9.** Run-up vulnerability indexes for the Lalzit Bay, using near-shore data for the 50 yr return period for different directional patterns:
A) 190-250°N sector; B) 250-350°N sector.

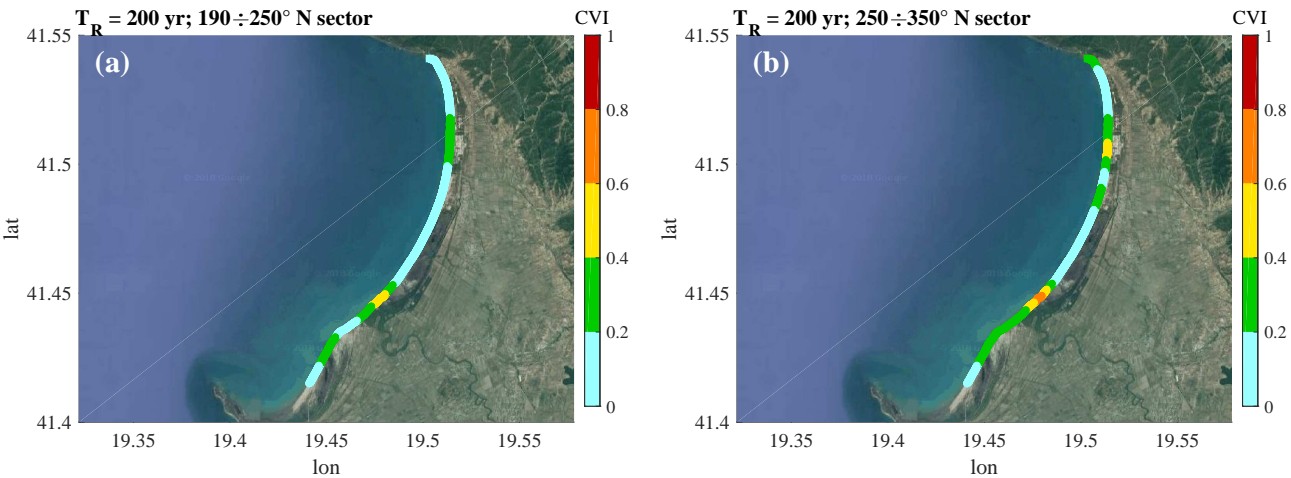

**Figure 10.** Run-up vulnerability indexes for the Lalzit Bay, using near-shore data for the 200 yr return period for different directional
patterns: A) 190-250°N sector; B) 250-350°N sector.

For the sake of clarity, in order to compare the results obtained with the two different approaches mentioned before, we
discuss just the results linked to the punctual investigated sections; analogous considerations can be therefore extended to the
intermediate sections, whose vulnerability levels were assessed through a linear interpolation as previously explained.

Looking at the punctual results (Fig. 11 and 12), it can be noticed that in all considered cases even sections lying next to
5   each other can show very different vulnerability levels: as the sampling locations are 1 km distant one from another, their
morphological characteristics can significantly vary, and this is consequently reflected in the results.



Referring to the regional scale-offshore analysis and 50 yr return period, the vulnerability towards inundation happens to be "very high" in section 0, and still "high" in sections 7, 8; sections -4, 1, 2 are characterized by a "very low" vulnerability, whereas sections 3, 4, 6, 9 and 10 show "low" vulnerability; the other ones are characterized by a "medium" vulnerability. As we could expect, vulnerability levels increase when referring to 200 yr return period: in this case, a "very high" vulnerability

5   characterizes section 7 as well, whereas the level increase from "medium" to "high" in section -1, and from "low" to "medium" in section 9; vulnerability class does not change for sections -4, -3, -2, and for sections between 0 and 6.

    The directional analysis indicates that results are less varying with respect to the return period: if we refer to the 190-250° N sector, 50 yr return period, vulnerability levels are "very low" for all sections but 7 and 8, which show "low" vulnerability, and 0 ("medium vulnerability"). Switching to the 200 yr return period, vulnerability rises from "very low" to "low" in sections -3,

10   -1, being unvaried in all the other ones. Results are slightly different for the 250-350° N sector: in this case, 50 yr vulnerability is "low" (instead of "very low") for sections -3, -2, -1, 5, 11; section 7 shows a "medium" instead of a "low" vulnerability. Again, results proportionally increase due to the considered return period: differences between the previous directional sector can be noted in section -2, 5, 11, being characterized by a "low" vulnerability rather than by a "very low" one; sections 0 and 7 are respectively "high" and "medium" vulnerable, whereas they are characterized by a one step lower vulnerability levels

15   compared to S-SW fetch.

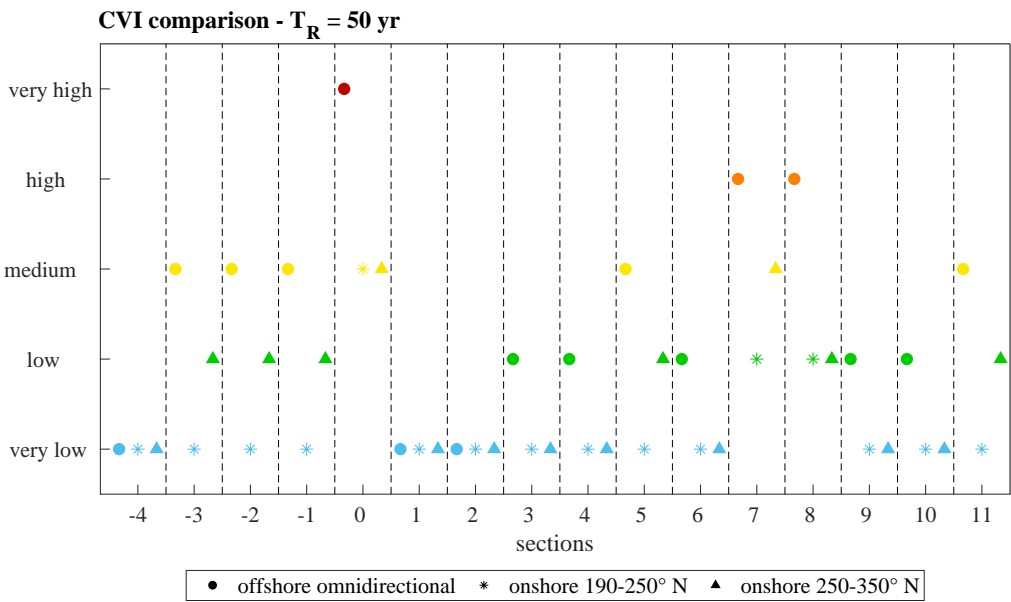

**Figure 11.** Comparison between the run-up vulnerability indexes for each sampling location; return period equal to 50 yr.





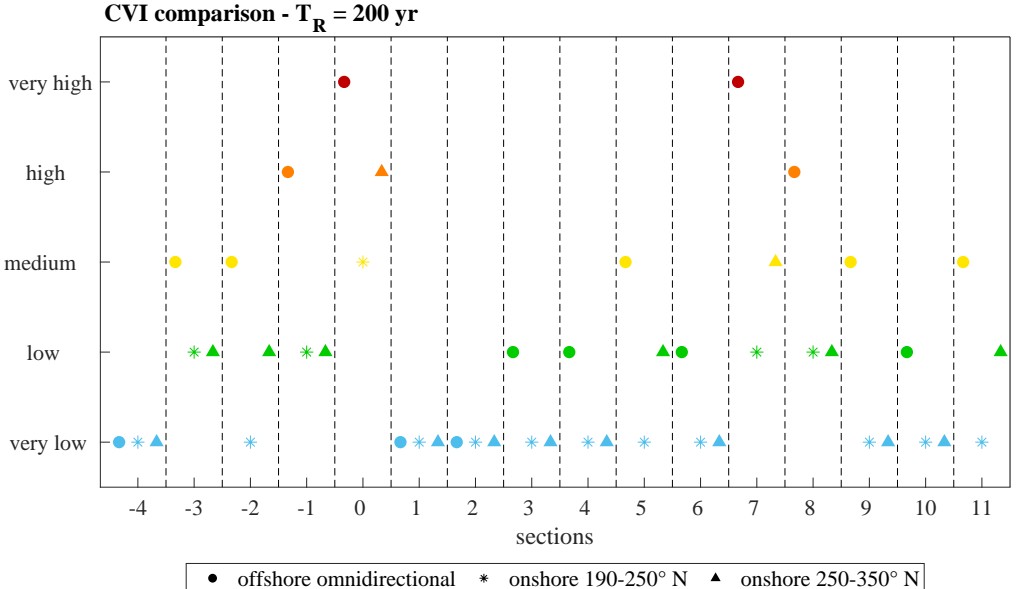

**Figure 12.** Comparison between the run-up vulnerability indexes for each sampling location; return period equal to 200 yr.

It is interesting to evaluate how $CVI$ can change due to the starting wave features: the extreme value analysis performed using deep-water data yields higher vulnerability levels than those obtained after propagating waves toward the shore. Referring to 50 yr return period, the most exposed sections are yet characterized by "very high" (0) and "high" (7, 8) levels of vulnerability, whereas is just considering the 200 yr return period that the directional analysis lead to a "high" level (section 0,

250-350° N sector); in this case, through omnidirectional analysis sections 7 and -1 become "very high" and "high" vulnerable respectively, whereas the directional analysis carries vulnerability levels never higher than "medium" but that of section 0, precisely.

Actually, results divergence decreases for sections characterized by "low" and "very low" vulnerability levels: in this case, the morphology of the surrounding beach seems to guarantee safe conditions, regardless to the magnitude of the forcing wave.

## 4    Discussion

As a general trend, assessing coastal vulnerability to inundation using the wave climate computed at the local scale leads to lower vulnerability levels compared to those obtained through the regional analysis. If the vulnerability levels are similarly distributed along the bay (depending on the single section profiles), the long term run-up estimates are clearly dependent to the referring spatial scale: the geometry of the bay indeed strongly affects the waves' propagation toward the coast. Moving

onshore, in the Lalzit bay wave heights likely decrease due to refraction and diffraction, which can be expected to be the dominant processes as suggested by the concave enclosed shape of the coast. Consequently, run-up estimates come to be lower when dealing with the local-scale analysis, and resulting $CVI$ behave accordingly. Results reported in Fig. 11 and 12 highlight




another important aspect: if we refer to the local scale, the vulnerability of the bay as a whole is higher when looking at the summer months (250-350°N). This outcome is justified as well by the geometry of the bay, in fact, even if the starting wave features for the different directional frames are similar (Table 2), waves coming from W-NW are not diffracted by the southern cape as it happens instead for those coming from S-SW. The absence of obstacles along the waves path implies a

lower reduction of the wave heights, involving in turn higher values of the following run-ups, thus higher values for the $IV$ variables (see Fig. 13A, B).

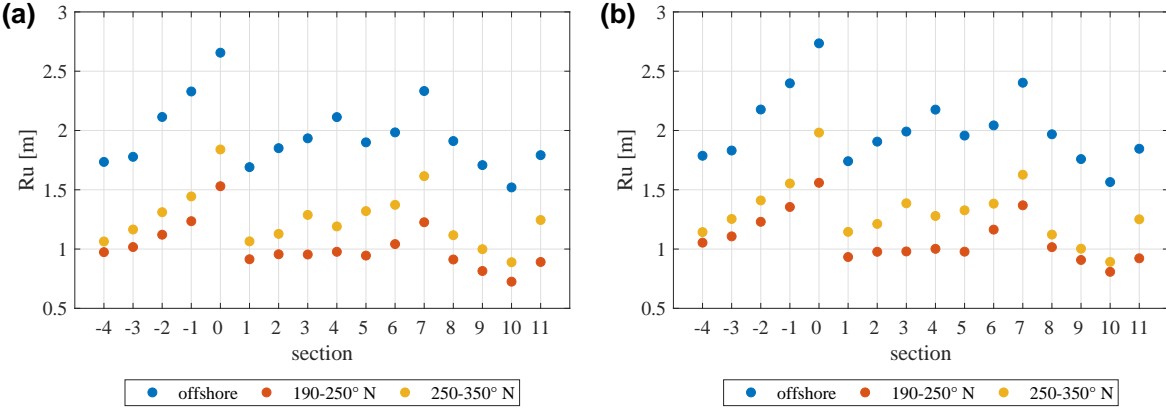

**Figure 13.** Comparison between run-up values for each section obtained through offshore (regional scale) and near-shore (local scale) conditions: A) $50yr$ return period; B) $200yr$ return period

Higher run-up estimates due to offshore analysis suggest another consideration about the different variability of the results between regional (offshore) and local (onshore) analysis: as previously demonstrated, the directional data result in a more homogeneous vulnerability level along the coastline. This can be simply justified looking at the vulnerability level computation:

the same $IV$ index may belong to different vulnerability classes, depending on the value that the $IV_{max}$ variable gets; in fact, while $IV_{min}$ is constant for any of the investigation approaches, the maximum $IV$ depends on the run-up values (see Eq. (2)). High run-up imply lower $IV_{max}$ values, thus a lower total range, which, being spaced in five classes anyway, leads to narrower intervals. Resulting vulnerability levels are therefore more sensitive to smaller variations of the $IV$ values (as Fig. 11 and 12 show).

Finally, if we enlarge our analysis to the coastline as a whole, we can better appreciate how vulnerability is distributed. Despite the differences due to the referring wave data, the most vulnerable areas happen to be those nearby the Erzeni outflow and, in the north, toward the Rodoni cape (see Fig. 4A for references), even if for different causes. If we look at the berm height component, it is evident how the aforementioned areas are characterized by lower berms (Fig. 4D): the Erzeni outflow area has shown in the last years a significant ongoing coastal erosion, as it is estimated that the coastline is retreating at a

speed of $0.3 \div 0.5$ m/y (Boçi, 1994), resulting in the berms levelling; actually, the concurring reduction of the river sediment transport has also implied steeper profiles (Fig. 4C), that lead to higher run-up estimates. Moving to the north, the lower berms





are due instead to the anthropic activities recently developed, which required the levelling of the beach as well; concerning the cross-shore slope, there is actually no evidence of steeper profiles but that of section 7.

## 5   Conclusions

The vulnerability assessment of a coastline can be a helpful device to plan its land use, as an instance not considering to place
high value activities when there's a high risk for the beaches of being submerged or eroded. In this framework, $CVI$ provide an easy and reliable tool, in order to get an overall overview about a shore vulnerability distribution toward either inundation and/or erosion events.

In this paper, we evaluated the coastal inundation vulnerability for the bay of Lalzit (Durrës, Albania), following the model proposed by Bosom García and Jiménez Quintana (2011). We first performed a regional analysis, referring to the original
formula of Stockdon et al. (2006) in order to compute the extreme values for the run-ups at sixteen sections along the bay; then, we detailed the study downscaling the wave features in the shallow waters thanks to a wave propagation model.

We showed that, even if the vulnerability distribution do not change along the shore (e.g. the most exposed sections are placed in the same areas), the results linked to the local scale yield considerably lower vulnerability levels. This is mainly due to the run-up estimates, which are very sensitive to the input wave characteristics, which may be defined in shallow or deep
waters. In the case of Lalzit, when waves propagation processes (such as refraction and breaking) become influential, run-up estimates can considerably change depending on the level of detail of wave characterization, as vulnerability levels accordingly do.

Since the model of Stockdon et al. (2006) quantifies extreme values for the run-up variable, it appears more plausible to refer to the modified model as proposed by Sancho-García et al. (2012), to link it with high return period, without computing
extreme analysis twice at a time. This precaution may allow to get more reliable $CVI$ assessment, properly scaling their related values due to the chosen return period. A critical analysis of the coastline vulnerability could prevent to adopt too much conservative approaches, that could lead to unnecessary countermeasures, translating to loss of money and invasive non required interventions.

The feasibility of $CVI$ assessment can represent a crucial ingredient for rapidly developing and transforming coastal regions
such as the Lalzit bay in Albania, which present more options to drive virtuous future coastal development compared to industrialized countries, where $CVI$ assessment may mostly represent a tool for ICZM applied to manage conflicts among relevant stakeholders.

*Acknowledgements.* This study is part of a project shared between the University of Trento and the University of Genoa (Italy), along with the Polytechnic of Tirana (Albania). The authors would like to thank everyone who joined the field data collection: Alessandro Chesini,
Alessandro Dotto, Alessio Maier, Daniele Spada, Dario Guirreri, Erasmo Vella, Federica Pedon, Giorgio Gallerani, Laura Dalla Valle, Martina Costi, Navarro Ferronato, Stefano Gobbi, Tommaso Tosi (University of Trento), Ardit Omeri, Arsela Caka, Bardhe Gjini, Bestar Cekrezi, Erida Beqiri, Ferdinand Fufaj, Idlir Lami, Marie Shyti, Mikel Zhidro, Nelisa Haxhi, Xhon Kraja, Tania Floqi (Tirana Polytechnic). The col-





lected data were then analysed by the Italian partners, in the framework of the UNESCO Chair in Engineering for Human and Sustainable Development (DICAM-Unesco Chair). G. Besio has been funded by University of Genoa through "Fondi per l'Internazionalizzazione" grant.





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
