# Peer review of "Coastal vulnerability assessment: through regional to local downscaling of wave characteristics along the bay of Lalzit (Albania)"

_Natural Hazards and Earth System Sciences, 2018_

## Referee Comment (RC1) · J. A. Jiménez (Referee) · 5 Jun 2018

Review of *Coastal vulnerability assessment: through regional to local downscaling of wave characteristics along the Bay of Lalzit (Albania)*

Manuscript **# nhess-2018-113**

The manuscript presents a vulnerability assessment to storm-induced inundation in an Albanian coastal stretch by using a vulnerability index. The topic is on the scope of NHESS and can be of interest for NHESS readers. In what follows, some comments are given.

**[1]** The first comment is purely formal. Although the index used by authors can formally be denoted as a coastal vulnerability index, I recommend authors to do not use CVI to refer to the index to avoid misunderstandings with readers since CVI is usually employed to name the Coastal Vulnerability Index developed by Gornitz et al (1994) and all variations used by USGS and others. Take this in consideration and apply throughout the manuscript.

**[2]** *Lines 14-18 (pag 2).* The interpretation of the runup model of Stockdon et al (2006) does not seem to be formally correct. The 2% factor is common to many of runup models (and, probably, the most used statistical value in flood-hazard assessments), and it refers the value exceeded by 2% of the runup values induced during a given wave-state (which is characterized by the use of $Hs$).In essence, the idea is to select a statistical description of the runup distribution (induced by the random wave state) and its selection will depend on the objective of the analysis. This 2% does not refer to probability of exceedance or return period as the text seems to suggest.

**[3]** *Lines 18-24 (pag 2).* This type of observation has also been previously done for other runup models. It is due to the fact that if the model is going to be fed with deepwater input data (as it is the case of most of runup models), if wave conditions significantly modify during propagation (diffraction, irregular bathymetry), used data will not properly represent real nearshore conditions. In any case and, just regarding the use of the Stockdon runup model, I include here two references that can be helpful.

Plant & Stockdon. (2015). How well can wave runup be predicted? Comment on Laudier et al.(2011) and Stockdon et al.(2006). *Coastal Engineering*, *102*, 44-48.
Stockdon et al (2014). Evaluation of wave runup predictions from numerical and parametric models. *Coastal Engineering*, *92*, 1-11.

**[4]** *Line1 (pag 3).* The right cite of Bosom García and Jiménez Quintana (2011) is Bosom and Jiménez (2011). Please change through the manuscript.

**[5]** Section 2.3 (pag 6). Please mention that this is (or it is based on) the index/method proposed by Bosom and Jiménez (2011).

**[6]** Lines 9-10 (pag 6). Please be explicit with the "physical meaning" of used intervals. (e.g something like this... *The minimum value correspond to a configuration in which the beach is not overtopped and, in consequence, the hinterland is well protected from inundation for the tested conditions. On the other hand, the maximum value ….*).

**[7]** *Lines 8-9 (pag 7).* With this approach you are assuming that the probability of the hazard (runup) is the same that the probability of the wave height. However, this is not exactly true since *Ru* depends on *Hs* and *Tp*. The strict way to obtain the 38-years time-series of annual maxima *Ru*, will be to compute *Ru* of all conditions during each year and to retain the maximum value every year.

**[8]** Line 14-15 (pag 7). The use of a given *Tr* is not conservative by itself. The appropriateness of the used value will depend on the safety level of the analysis (which should be related to the value of the hinterland potentially exposed to inundation).

**[9]** *Lines 21 to 30 (pag 7).* Here I have some doubts about the authors' approach. If the objective is to do a **seasonal analysis**, the approach is easier than the used by authors. Essentially will be to split each original time series into *N* series, where *N* is the number of seasons to be used in the analysis (2, 3, 4) and, then, to apply GEV (as it was previously done with the total time-series) to each seasonal-representative annual maxima *Ru* time series. Of course, if authors want to use nearshore data, offshore conditions need to be propagated towards the coast. So with this, direction is implicit to the analysis since each seasonal data set will only include waves corresponding to such season, and if there is any directional preference, this will be reflected in the analysis.

**[10]** *Line 1 (pag 8).* If **[9]** is applied this is not true, you will obtain N (being N the number of seasons) time series of annual maxima with the same data number than using the total time-series.

**[11]** *Lines 2-4 (pag 8).* A threshold value of *Hs* 3 m seems to be high for the area to apply POT. Why authors used this value? How many average storms per year do you obtain? To which percentile of the cumulative distribution is equivalent this value?

**[12]** *Line 8 (pag 8).* The use of the empirical formula of Callaghan et al (2008) is quite problematic and, probably, not directly applicable. The formula (used coefficients) was obtained with data from a wave buoy in Botany Bay (Australia) where conditions are expected to be substantially different to the one sin the study area.

**[13]** *Lines 14-15 (pag 8).* This will depend on the characteristics of the wave climate of the study site (see comment [7]).

**[14]** *Section 3.2* Results presented here cannot formally be named as seasonal analysis but directional analysis. As the text indicates *(line 14, pag 10)* authors divide wave sin directions and apply EVA to each (directional) dataset.

**[15]** *Pag 11.* Results showing T calculated in function of H using eq 5 are not valid (see comment 13). Coefficients of eq 5 have to be derived from local data.

**[16]** *Results, Discussion and Conclusions* maybe affected by previous comments. Adapt these sections once you decide on them.

**[17]** *Section 5.* The comment that using deepwater or nearshore waves give a more reliable CVI assessment is not necessarily true. This will permit to use a wave height more representative of local wave conditions. But, the vulnerability assessment will be more robust or valid provided that CVI properly reflect the conditions of the area. To validate this, you need to compare calculations with reality (e.g. are the vulnerable area usually affected by inundation?).

**Formal issues**

**Fig [1]** Please combine Figs 1 (need to be improved) and 2 (also to be improved) in just one figure.

**References.** Please check them carefully. Some of them are incomplete (e.g. Armaroli and Duo, 2017; Battjes'71), badly cited (De Leo et al. 2017), authors bad included (Bosom and Jiménez 2011; Oscar Ferreira et al. 2017).

Please check the grammar through the manuscript.

---

## Referee Comment (RC2) · Anonymous Referee #2 · 23 Jul 2018

This paper deals with coastal vulnerability assessment at a coastal area in Albania. The topic of this article is relevant for the Journal and for the Scientific audience, even more when climate change will have a strong impact worldwide.

Despite the title of the article appears to be a case study, after reading the manuscript it is focused on the enhancement of one of the key aspects that should be carefully study in the near future: global or regional scenarios must be properly transform to the study site if we want to achieve relevant and close to reality results and conclusions. This transformation depends, among other things, on the quality of the data, and in some cases this information is not available.

[Figure]

The paper is written clearly and the formulations and methodology are up-to-date, with particular emphasis on the propagation of the wave climate. The figures are of nice quality and results and discussion sections are written clearly. I found the discussion section very enlightening. I only suggest revising the references (i.e. line 7 "Oscar Ferreira").

---

## Author Comment (AC1) · 20 Aug 2018

Dear Editor,

we would like to thank the reviewers for their effort in reading and commenting the paper. We went through all the comments and we tried to answer in detail to all of them. An item-by-item reply follows for the Reviewer 1 revision. For those comments on which we could not agree, a detailed rebuttal is presented in the reply which follows.

**REVIEWER 1**

We thank the Reviewer for his interest in our paper and for his criticism. We have considered all the review comments and undertaken a major revision, incorporating nearly all the requested changes into the manuscript.
* * *
[1] *The first comment is purely formal. Although the index used by authors can formally be denoted as a coastal vulnerability index, I recommend authors to do not use CVI to refer to the index to avoid misunderstandings with readers since CVI is usually employed to name the Coastal Vulnerability Index developed by Gornitz et al. (1994) and all variations used by USGS and others. Take this in consideration and apply throughout the manuscript.*

We recognize that the notation "CVI" was originally defined for the index of Gornitz et al. (1994) and its variations; actually, we were keeping this notation as there are other studies presenting the development of Coastal Vulnerability Indexes literally named as CVI, even though the proposed methodology does not precisely follow that of Gornitz et al. (1994) (see, as an instance, Kumar et al., 2010; McLaughlin et al., 2002, among others)

Anyway, in order to avoid any possible misunderstanding, we decided not to longer use the notation "CVI", speaking instead of "vulnerability level" (or "VL") of the coast. We therefore changed it throughout the whole paper.
* * *
[2] *Lines 14-18 (pag 2). The interpretation of the runup model of Stockdon et al. (2006) does not seem to be formally correct. The 2% factor is common to many of runup models (and, probably, the most used statistical value in flood-hazard assessments), and it refers the value exceeded by 2% of the runup values induced during a given wave-state (which is characterized by the use of Hs). In essence, the idea is to select a statistical description of the runup distribution (induced by the random wave state) and its selection will depend on the objective of the analysis. This 2% does not refer to probability of exceedance or return period as the text seems to suggest.*

We acknowledge that the explanation is a little bit confusing; clearly the return period of an event is linked to its probability of non-exceedance among the distribution it may belong to, but this applies in the frame of EVA. In this case, the 2% refers instead to the distribution of a single sea state, so it cannot be linked to any return period, as the referee correctly pointed out. However, this percentage is actually tied to a probability of exceedance, as "it refers to the value exceeded by 2% of the runup values induced during a given wave-state" (according to the referee's comment); thus, it represents a high quantile among a hypothetical series of observed runup values during a sea state. Through the propagation of the waves at a 10m depth, we are referring instead to an *expected value* of a sea state induced run-up (as proved

by Sancho-García et al., 2012), to see how a different approach may affect the computation of the resultant vulnerability levels.

The discussion has been changed in lines 13-20 of pag.2, in order to better clarify the differences between the two approaches.
* * *
[3] *Lines 18-24 (pag 2). This type of observation has also been previously done for other runup models. It is due to the fact that if the model is going to be fed with deepwater input data (as it is the case of most of runup models), if wave conditions significantly modify during propagation (diffraction, irregular bathymetry), used data will not properly represent real nearshore conditions. In any case and, just regarding the use of the Stockdon runup model, I include here two references that can be helpful.*
*Plant & Stockdon. (2015). How well can wave runup be predicted? Comment on Laudier et al. (2011) and Stockdon et al.(2006). Coastal Engineering, 102, 44-48.*
*Stockdon et al (2014). Evaluation of wave runup predictions from numerical and parametric models. Coastal Engineering, 92, 1-11.*

We appreciate the suggestions.

As regard the paper of Plant and Stockdon (2015), we precise that we used the full parametrization of Stockdon et al. (2006) as the authors suggest; moreover, the beach of the Lalzit bay is barred, so no hypothetical additional sources of uncertainty may exist (as Laudier et al., 2011, suggested comparing the Stockdon et al. (2006) model performances on non-barred beaches).

Looking instead at Stockdon et al. (2014), a comparisons for swash and setup values between Stockdon et al. (2006) and the XBeach simulations for "extreme conditions" was carried out, in order to evalute the applicability of Stockdon et al. (2006) for very intense sea states: results were consistent for infragravity swash, while for the setup they agree just for the category I storm (defined according to the Saffir-Simpson scale). Nevertheless, in the Adriatic Sea, climate is never characterized by winds belonging to the higher categories, so that we can use this reference to justify the use of Stockdon et al. (2006) for the local extreme sea states of the bay of Lalzit.

Finally, according to the references proposed by the reviewer, we are referring to the Stockdon et al. (2006) parametrization as "S2006".
* * *
[4] *Line1 (pag 3). The right cite of Bosom García and Jiménez Quintana (2011) is Bosom and Jiménez 2011). Please change through the manuscript.*

Thank for pointing it out (we used the pre-defined settings of the Copernicus package as suggested by the Journal). The typo has been fixed throughout the paper.
* * *
[5] *Section 2.3 (pag 6). Please mention that this is (or it is based on) the index/method proposed by Bosom and Jiménez (2011).*

Amended as suggested.
* * *
[6] *Lines 9-10 (pag 6). Please be explicit with the "physical meaning" of used intervals. (e.g something like this... The minimum value corresponds to a configuration in which the beach is not overtopped and, in consequence, the hinterland is well protected from inundation for the tested conditions. On the other hand, the maximum value...).*

Amended as suggested: the meaning of the term is explained at line 10, pag. 6.
* * *
[7] *Lines 8-9 (pag 7). With this approach you are assuming that the probability of the hazard (runup) is the same that the probability of the wave height. However, this is not exactly true since Ru depends on Hs and Tp. The strict way to obtain the 38-year time-series of annual maxima Ru, will be to compute Ru of all conditions during each year and to retain the maximum value every year.*

We agree on that; we used this analysis to ensure a coherent comparison with the results due to the propagation of waves towards the coast. In this last case, according to the referee comment, we would have had to compute the long term run-up starting from the whole series of propagated waves parameters; nevertheless, to downscale 38 years of hourly defined wave data may be computationally too expensive. Therefore, when dealing with the off-shore analysis, we first evaluated the 38 maxima runup due to the "strict way" as explained in the referee's comment; then, we compared these values with those computed according to the procedure explained in the paper (lines 8-9, pag.7). This analysis was performed for each of the sixteen locations, the results for the less steep and the steepest sections are shown in Figure 1 of the present reply.

[Figure]

Figure 1: Yearly maxima run-up comparison for the sections characterized by the minimum and the maximum slope (10 and 0, respectively).

Actually, the two approaches for run-up computation lead to similar results, and this is reflecting on the long-term curves which happen to lie very close to each other; therefore,

resultant vulnerability levels are not sensitive to the selected approach for our case study. Then, even though there is no evidence that the same would occur with the directional on-shore analysis, there is no reason for us to expect huge divergences. Moreover, the same kind of analysis has been previously performed in other papers, as reported in the manuscript (lines 14-17, pag.8). We decided not to put these results for the sake of breavity, nonetheless we are commenting them at the end of line 9, pag.7.
* * *
[8] *Line 14-15 (pag 7). The use of a given Tr is not conservative by itself. The appropriateness of the used value will depend on the safety level of the analysis (which should be related to the value of the hinterland potentially exposed to inundation).*

We agree with the reviewer that a referring return period has to be defined according to a required safety level. Here, beside the characterization of the vulnerability levels for the bay of Lalzit, we want to show how to rely on a regional or a local scale may affect these levels themselves. For a more meaningful overview, we referred to two different return periods, explanative of two different hypothetical safety levels: that's the reason for that we talk about "differently conservative approaches".
* * *
[9] *Lines 21 to 30 (pag 7). Here I have some doubts about the authors' approach. If the objective is to do a seasonal analysis, the approach is easier than the used by authors. Essentially will be to split each original time series into N series, where N is the number of seasons to be used in the analysis (2, 3, 4) and, then, to apply GEV (as it was previously done with the total time-series) to each seasonal-representative annual maxima Ru time series. Of course, if authors want to use nearshore data, offshore conditions need to be propagated towards the coast. So with this, direction is implicit to the analysis since each seasonal data set will only include waves corresponding to such season, and if there is any directional preference, this will be reflected in the analysis.*

The original goal of the paper is not to perform a seasonal analysis, but a directional one according to the waves' main incident directions of the local wave climate. Actually, we agree that it would be easier to refer to the annual maxima approach (AM) even when performing the directional analysis; moreover, that would ensure as well a more coherent approach to that used for the omnidirectional vulnerability levels computation. We therefore decided to switch from POT to AM also for the computation of the on-shore vulnerability levels. Moreover, we redefined the boundaries of the directional sectors in order to better group the annual maxima $H_s$ incident directions within every single sector. The new sectors are defined as follows:

- Sector 1: $157.5 - 247.5°N$ corresponding to the sectors of Mezzogiorno and Libeccio winds

- Sector 2: $247.5 - 337.5°N$ corresponding to the sectors of Ponente and Mistral winds

In any case, a directional analysis requires to compute the probabilities according to Eq. (4) as explained in the paper.

If we look at the roses of Fig. 7 in the paper (pag. 10), it can be noticed how the waves are most likely to come from a directional sector according to given periods of the year.

Nevertheless, when looking to the AM data, this trend is no longer evident, event though a less marked seasonality for the waves belonging to Sector 2 can be still appreciated (see figure 2 of the present reply). We therefore agree that speaking of "seasonality" for the AM data may confuse the reader, for that we are reviewing the terms "seasonal" and "seasonality" through the whole manuscript, to make clearer the main focus of the investigation.

[Figure]

Figure 2: Seasonality of the yearly maxima $H_s$ belonging to the different directional sector.
* * *
[10] *Line 1 (pag 8). If [9] is applied this is not true, you will obtain N (being N the number of seasons) time series of annual maxima with the same data number than using the total time-series.*

As explained in comment [9], we are performing a directional analysis, and we switched to an AM approach. Then, we are obtained N number of 38 data (being 38 the number of years the hindcast data is defined on), said N the number of directional sector considered.
* * *
[11] *Lines 2-4 (pag 8). A threshold value of Hs 3 m seems to be high for the area to apply POT. Why authors used this value? How many average storms per year do you obtain? To which percentile of the cumulative distribution is equivalent this value?*

The value of 3m correspond to the 99.2% quantile of the initial $H_s$ distributions. We retained this threshold after performing a sensitivity analysis on the threshold values, given an inter-event duration of 24hours. The value of 3m ensured the resulting dataset to be i.i.d. according to the Kendall's test (Ferguson et al., 2000) and Poisson distributed in frequency due to the Fisher's statistic (Ferguson et al., 2000). Anyway, switching to the AM approach for the directional analysis too, we are no longer commenting on the threshold's selection (thus reviewing all the lines 1-7, pag.8 in the manuscript).
* * *
[12] *Line 8 (pag. 8). The use of the empirical formula of Callaghan et al (2008) is quite problematic and, probably, not directly applicable. The formula (used coefficients) was obtained with data from a wave buoy in Botany Bay (Australia) where conditions are expected to be substantially different to the one in the study area.*

We definitively agree that the empirical model of Callaghan et al. (2008) has to be tested over the local wave climate conditions. Actually, we selected this formula as, for the strongest sea states of the referring hindcast location, it performs better than the other empirical models commonly encountered in literature (e.g Goda, 2003; Boccotti, 2004, see Fig. 3 of the present reply).

[Figure]

Figure 3: Fitting of different empirical models to the wave height and period of Point_002550.dat dataset.
* * *
[13] *Lines 14-15 (pag 8). This will depend on the characteristics of the wave climate of the study site (see comment [7]).*

We agree with the referee: "not necessarily" (line 14, pag.8) was actually written to take into account this possibility; in order to enhance this point, we are adding the proposed comment at the end of the line 15, pag. 8. Anyway, looking at the local wave climate, we can reasonably assume that the return periods for wave height and runup lead to very close probability of exceedance (see comment [7]).
* * *
[14] *Section 3.2 Results presented here cannot formally be named as seasonal analysis but directional analysis. As the text indicates (line 14, pag 10) authors divide wave sin directions and apply EVA to each (directional) dataset.*

As we explained in comment [9], we performed the analysis dividing the incoming wave climate due to two directional patterns. Actually, even when presenting the results of Fig. 9-10-11-12-13 in the paper, we speak of "directional sectors". Nevertheless, as the aforementioned patterns happened to show different seasonal dependences, we considered appropriate to spend a comment on it, even if we did not statistically characterize this link. As previously stated, we reviewed the adopted terminology in order to avoid any misunderstanding.
* * *
[15] *Pag 11. Results showing T calculated in function of H using eq 5 are not valid (see comment 13). Coefficients of eq 5 have to be derived from local data.*

As we explained in comment [12], although the model of Callaghan et al. (2008) was validated for a different wave climate than that of the area under investigation, we referred to it as it happened to better models the most energetic events. As explained in comment [9], we considered to incorporate the referee's suggestion to adopt an AM approach even for the directional analysis. Thus, we performed again the propagation of waves towards the coast, defining, for each sector at a time, the input wave parameters as follows:

- as regards the significant wave height we selected the data through an AM approach, modelling at a second time through a GEV distribution for the long term parameter estimation;

- regarding the peak period, we are still referring to the model of Callaghan et al. (2008), justifying this choice as explained in [12];

- the incoming direction is still defined according to the approach explained in the paper (Sect. 3.2, pag. 11), with the only difference that the directions are those of the AM events, no longer the POT ones.
* * *
[16] *Results, Discussion and Conclusions maybe affected by previous comments. Adapt these sections once you decide on them.*

We reviewed Sections 3, 4 and 5 due to the changes done amending reviewer's comments.
* * *
[17] *Section 5. The comment that using deepwater or nearshore waves give a more reliable CVI assessment is not necessarily true. This will permit to use a wave height more representative of local wave conditions. But, the vulnerability assessment will be more robust or valid provided that CVI properly reflect the conditions of the area. To validate this, you need to compare calculations with reality (e.g. are the vulnerable area usually affected by inundation?).*

We agree that a comparison with historic records would be the key to prove a better vulnerability estimation; unfortunately, as far as we know, this dataset does not exist (or it is not accessible). We spoke of "reliability" as, being the CVI computation closely tied to the incoming wave climate, a proper characterization of it may consequently better detail the vulnerability of the investigated area. This is especially true when the geometry of the bay affects the propagation of the waves towards the shore, as previously noticed in comment [3].

In Section 5, we reviewed lines 18-23 in order to avoid misunderstandings: firstly, on the term "extremes analysis" (as described in comment [2]); secondly, on the use of the term "more reliable", as just explained.

**Formal issues**

- *Fig [1] Please combine Figs 1 (need to be improved) and 2 (also to be improved) in just one figure.*
  We are removing Fig. 2 as we agreed that it was redundant. We are changing the format of Fig. 1a) to make it clearer.

- *References. Please check them carefully. Some of them are incomplete (e.g. Armaroli and Duo, 2017; Battjes'71), badly cited (De Leo et al. 2017), authors bad included (Bosom and Jiménez 2011; Oscar Ferreira et al. 2017).*
  Amended as suggested.

- *Please check the grammar through the manuscript.*
  Amended as suggested.

**References**

Boccotti, P. (2004). *Idraulica marittima*. Utet.

Callaghan, D., P. Nielsen, A. Short, and R. Ranasinghe (2008). Statistical simulation of wave climate and extreme beach erosion. *Coastal Engineering 55*(5), 375–390.

Ferguson, T. S., C. Genest, and M. Hallin (2000). Kendall's tau for serial dependence. *Canadian Journal of Statistics 28*(3), 587–604.

Goda, Y. (2003). Revisiting wilson's formulas for simplified wind-wave prediction. *Journal of waterway, port, coastal, and ocean engineering 129*(2), 93–95.

Gornitz, V. M., R. C. Daniels, T. W. White, and K. R. Birdwell (1994). The development of a coastal risk assessment database: vulnerability to sea-level rise in the us southeast. *Journal of Coastal Research*, 327–338.

Kumar, T. S., R. Mahendra, S. Nayak, K. Radhakrishnan, and K. Sahu (2010). Coastal vulnerability assessment for orissa state, east coast of india. *Journal of Coastal Research*, 523–534.

Laudier, N. A., E. B. Thornton, and J. MacMahan (2011). Measured and modeled wave overtopping on a natural beach. *Coastal Engineering 58*(9), 815–825.

McLaughlin, S., J. McKenna, and J. Cooper (2002). Socio-economic data in coastal vulnerability indices: constraints and opportunities. *Journal of Coastal Research 36*(sp1), 487–497.

Plant, N. G. and H. F. Stockdon (2015). How well can wave runup be predicted? comment on laudier et al.(2011) and stockdon et al.(2006). *Coastal Engineering 102*, 44–48.

Sancho-García, A., J. Guillén, G. Simarro, R. Medina, and V. Cánovas (2012). Beach inundation prediction during storms using direferents wave heights as inputs. International Conference on Coastal Engineering.

Stockdon, H., R. Holman, P. Howd, and A. Sallenger Jr. (2006, May). Empirical parameterization of setup, swash, and runup. *Coastal Engineering 53*(7), 573–588.

Stockdon, H. F., D. M. Thompson, N. G. Plant, and J. W. Long (2014). Evaluation of wave runup predictions from numerical and parametric models. *Coastal Engineering 92*, 1–11.

---

## Author Comment (AC2) · 20 Aug 2018

Dear Editor,

we would like to thank the reviewers for their effort in reading and commenting the paper. We went through all the comments and we tried to answer in detail to all of them. An item-by-item reply follows for the Reviewer 2 revisions.

**REVIEWER 2**

*The paper is written clearly and the formulations and methodology are up-to-date, with particular emphasis on the propagation of the wave climate. The figures are of nice quality and results and discussion sections are written clearly. I found the discussion section very enlightening. I only suggest revising the references (i.e. line 7 "Oscar Ferreira").*

We thank the Reviewer for his/her interest in our paper and the comments he/she spend on it; as regards the references, we manually fixed the typos present in the paper due to the automatic settings of the Copernicus package, as the referee suggested.

---

## Referee Report (RR1)

Review of ***Coastal vulnerability assessment: through regional to local downscaling of wave characteristics along the Bay of Lalzit (Albania)***

Manuscript **# nhess-2018-113-Rev1**

Authors have satisfactorily addressed most of reviewer's comments and suggestions done to the original version of the manuscript. In spite of this, I have two major remarks and two minor observations:

**[1]** Authors have decided to maintain the use of the empirical formula of Callaghan et al (2008) to characterize the relationship between T and H at the study area. This is justified in basis of a better performance than other empirical models commonly encountered in the literature. This is NOT a good justification because the best empiric model should be that directly obtained from your own data. Since authors have the original data (time series of H, T, direction) why do not obtain such relationship by fitting a given function for your data (for those during extreme events, i.e. when Hs > 3 m). This would be the best empiric relationship you can get for your case.

Since T is affecting runup magnitude and also wave propagation, this is a critical issue in your analysis and must be properly justified.

**[2]** When doing the local scale analysis (section 3.2), authors propagate selected event to the coast to account for changes in wave conditions and, in consequence, in Ru due to existing bathymetry. This is the usual approach when we want to assess the effect of an irregular bathymetry but, it is incomplete. The usual way to do this is, once wave are propagated towards the coast over the real bathymetry using a wave propagation model, obtained values are propagated backward using Snell law to obtain equivalent deepwater wave characteristics to feed runup model. This is the way to obtain coherent values to be consistently compared with your first computation (directly using deep water values).

Moreover, it should be great if you give some basic details on wave propagation (which is the used model?).

**Minor observations**
***Conclusions***
It will be more appropriate to name Chapter 5 as "Summary and Conclusions" since this reflects better its current content.
Remove in this section "(referred to as S2006)". This was already mentioned and now you can simply use it.

***References***
***Wrong citation***
De Leo et al. 2016. Must be *Proc 35th International Coastal Engineering Conference*, ASCE,
***Incomplete (no journal name included)***
Ferreira Silva et al. 2017
Sancho-García et al. 2012

---

## Author Response (AR2)

UNIVERSITÀ DEGLI STUDI DI GENOVA

**DICCA**

**D**ipartimento di **I**ngegneria **C**ivile, **C**himica e **A**mbientale

16145 GENOVA - Via Montallegro, 1 - Tel.  39 - 010 3532491 - Fax 39 - 010 3532546

The following document reports a step by step reply to the last review, along with the modified version of the paper.

The parts removed are in red, while the new ones are written in blue.

[Figure]

Dear Editor,

we would like to thank the reviewer for further reading and commenting the paper. We went through the comments and we answerd in detail to all of them. An item-by-item reply follows for the revision. For those comments on which we could not agree, a detailed rebuttal is presented in the reply which follows.

*[1] Authors have decided to maintain the use of the empirical formula of Callaghan et al (2008) to characterize the relationship between T and H at the study area. This is justified in basis of a better performance than other empirical models commonly encountered in the literature. This is NOT a good justification because the best empiric model should be that directly obtained from your own data. Since authors have the original data (time series of H, T, direction) why do not obtain such relationship by fitting a given function for your data (for those during extreme events, i.e. when Hs > 3 m). This would be the best empiric relationship you can get for your case. Since T is affecting runup magnitude and also wave propagation, this is a critical issue in your analysis and must be properly justified.*

We agree with the reviewer that the best relationship should be carried out directly from the original data. We therefore fit the log-normal model as proposed by Callaghan et al (2008) over the sea storms Hs/Tp of the directional sectors taken into account. We performed a PDS data selection, fixing the threshold at the 98 Hs' quantile and an inter-event duration of 24 hours, retaining Hs and Tp and estimating the best set of coefficients for the Callaghan's equation at a time.

Results are reported in Table 1; a comparison of the following long term periods is shown in Tables 2 and 3. It can be noticed that results happen to be very close, implying that neither the propagations of waves nor the final VL estimates are significantly affected.

However, we decided to rely on these new estimates to uniquely link the model to our case study; we are consequently modifying the explanation at page. 10 (see equation 5).

|   | Callaghan | Empirical - sector 1 | Empirical - sector 2 |
|---|---|---|---|
| a | 3.005 | 4.3819 | 5.0359 |
| b | 0.543 | 0.4134 | 0.4252 |
| c | 4.82 | 0.6815 | 3.8330 |
| d | -0.332 | 0.0766 | -1.8605 |
| f | 1.122 | 0.9875 | 3.1491 |
| g | -0.039 | 0.3368 | -1.6310 |

*Table 1: Callaghan coefficients*

| TR | Hs [m] | Tp Callaghan [s] | Tp emp [s] |
|---|---|---|---|
| 50 | 6.3 | 10.8957 | 10.8183 |
| 500 | 7.0 | 11.2717 | 11.2999 |

*Table 2: Long term period. Sector 1*

| TR | Hs [m] | Tp Callaghan [s] | Tp emp [s] |
|---|---|---|---|
| 50 | 5.6 | 10.5120 | 10.5057 |
| 500 | 6.0 | 10.7322 | 10.8113 |

*Table 3: Long term period. Sector 2*

*[2] When doing the local scale analysis (section 3.2), authors propagate selected event to the coast to account for changes in wave conditions and, in consequence, in Ru due to existing bathymetry. This is the usual approach when we want to assess the effect of an irregular bathymetry but, it is incomplete. The usual way to do this is, once wave are propagated towards the coast over the real bathymetry using a wave propagation model, obtained values are propagated backward using Snell law to obtain equivalent deepwater wave characteristics to feed runup model. This is the way to obtain coherent values to be consistently compared with your first computation (directly using deep water values).*
*Moreover, it should be great if you give some basic details on wave propagation (which is the used model?).*

The propagation of waves over the local bathymetry was aimed at evaluating how the wave parameters could be modified when estimated in the shallow-water area. In order to do that, we used the SWAN model, as mentioned in line 16 of page 7; we are commenting it more exhaustively but we would prefer not to explain the model in details (this is left to the proper reference of Booij et al 2003).
Sancho-García et al. (2012) proved that, feeding the equation of Stockdon et al (2006) with the parameters at a depth of 10m, provided run-up estimates closer to the ones they observed, this is the reason for we are talking of run-up estimates closer to an *expected value* (page 2, line 14).
To recap: S2006 may lead to a 2% run-up or to a XX% run-up (where XX is the percentage linked to the mean, it could be 50 if mean and median coincided), depending on the depth the referring waves parameters are defined at (off-shore/onshore, precisely).
Run-up is the key parameter for VL computation, so we think it would be interesting to see how different approaches may affect the final VL estimates; therefore, for our purpose we shall not to propagate the waves backward.
Actually, the computational scheme is similar to that of Vitousek et al (2008) (the "3$^{rd}$ approach" in their paper, cited at page 7 of our manuscript), where they either did not perform the backward propagation.

**Minor observations**

*Conclusions*
*It will be more appropriate to name Chapter 5 as "Summary and Conclusions" since this reflects better its current content. Remove in this section "(referred to as S2006)". This was already mentioned and now you can simply use it.*

Amended as suggested.

We fixed the citations as suggested.

**FURTHER COMMENT BY THE AUTHOR**
We modified Figures 5, 8 and 9 improving their resolution in order for the results to be clearer.

[revised manuscript text omitted]

---

## Author Response (AR3)

The following document reports the paper revised due to the editor suggestions.

The added paragraph is underlined in blue in the "Discussion" section.

[revised manuscript text omitted]